# The Effect of Perceived Real-Scene Environment of a River in a High-Density Urban Area on Emotions

**Mengyixin Li [1,*,†], Rui Liu [2,3,†], Xin Li [1,*], Shiyang Zhang [4] and Danzi Wu [4]**

[1] School of Architecture and Urban Planning, Beijing University of Civil Engineering and Architecture, Zhanlanguan Road 1, Xicheng District, Beijing 100044, China

[2] School of Geomatics and Urban Spatial Information, Beijing University of Civil Engineering and Architecture, Beijing 102616, China

[3] Key Laboratory of Urban Spatial Information, Ministry of Natural Resources, Beijing University of Civil Engineering and Architecture, Beijing 102616, China

[4] School of Landscape Architecture, Beijing Forestry University, Beijing 100083, China

[*] Correspondence: limengyixin@bucea.edu.cn (M.L.); lixin2@bucea.edu.cn (X.L.)

[†] These authors contributed equally to this work.

**Abstract:** Public sub-health has emerged as a pressing concern in densely populated urban areas. The urban environment, with its innate ability to modulate public emotions, harbors a precious resource in the form of urban rivers, which provide a serene and verdant space. This study focuses on the Liangma River in Chaoyang District, Beijing, selecting two rivers with diverse landscape features as the subjects of research. By employing physiological feedback data in conjunction with a subjective questionnaire, the emotional impact of high-density urban riverside spaces on individuals is quantitatively analyzed. Electrocardiogram (ECG) data, eye movement data, and the positive–negative emotion scale (PANAS) are subjected to data analysis. The study reveals the following key findings: (1) The riverside landscape in high-density urban areas exerts a positive influence on emotional well-being. Individuals in more natural river settings experience greater levels of contentment and relaxation, while those in areas with a higher proportion of artificial elements exhibit increased excitement and happiness. Moreover, scenes characterized by a greater degree of greening have a more pronounced soothing effect on mood. (2) A specific correlation between visual characteristics and emotional fluctuations is observed. The waterfront side of the trail exerts a stronger spatial attraction, and a higher proportion of blue and green spaces significantly contributes to stress relief. (3) The utilization of human-induced engineering technology, which captures emotional changes through physiological feedback, demonstrates a higher level of accuracy and is well-suited for small-scale studies. These findings highlight the potential of arranging diverse types of waterfront footpath landscapes in high-density urban areas and approaching waterfront landscape design and transformation from a novel perspective centered on health intervention. Such efforts hold promise for alleviating the daily pressures faced by the general public and fostering the development of a "healthy city".

**Keywords:** high-density city; riverside footpath; real-scene environment perception; physiological feedback; landscape preference

## 1. Introduction

With the burgeoning population and the influx to metropolitan areas, high density has become an inherent characteristic of numerous expansive cities in China [1]. According to the data disseminated by the Seventh National Population Statistics in 2021, Beijing's population density stood at 1334 people/km² [2], epitomizing the attributes of a high-density metropolis. The rapid economic advancement, unwavering efficacy, and multifaceted nature of the city's epicenter have magnetized an influx of inhabitants and

resources. Simultaneously, owing to the scarcity of land resources and the preponderance of complex landscape infrastructure, such as towering buildings and thoroughfares, the availability of premium green spaces is severely constrained. In addition, the waterfront areas are adversely affected by the towering structures and the varying characteristics of the high-density space environment, such as the visual intersection in the urban expanse. In light of the increasing population, the issue of public health has emerged as an exigent predicament confronting high-density urban agglomerations [3]. Therefore, the necessity of balancing urban development with the construction of verdant environments has taken center stage, with a focus on offering opportunities for denizens to engage with nature in high-density cities [4], while concurrently mitigating the deleterious repercussions of high-density surroundings on emotional well-being [5,6].

The early studies pertaining to the perception of emotions primarily relied on photographs, cognitive maps, environmental simulations, and other methodologies that employed subjective human descriptions and external behavior [7–9]. However, these approaches suffered from certain inaccuracies and hysteresis [10]. By employing physiological instruments, researchers are able to capture and observe real-time physiological signals, subsequently utilizing computer signal processing and analysis techniques to extract and categorize various emotions. This approach enables the accurate interpretation of neural processes and physiological changes in individuals, facilitating comparative analyses in conjunction with psychological indicators, and has thus emerged as the predominant research methodology [11–13]. Physiological signals such as blood volume pulse (BVP), electromyogram (EMG), electroencephalogram (EEG), electrocardiogram (ECG), electrodermal activity (EDA), salivary cortisol (S.C.), glucose (GLU), electrocardiogram(ECG), and face recognition measurement have been extensively employed in studies examining the natural environment, spatial characteristics of natural environment, urban parks, and the impact of plant landscape composition on human health and well-being [14–19]. Among the various physiological signals, ECG signals possess the advantages of real-time data acquisition, ease of collection, and high accuracy in emotion recognition [20]. In comparison to traditional questionnaire surveys, the utilization of physiological signals for emotion recognition is not susceptible to subjective influences, thereby enabling accurate research results even with a limited number of participants [21,22]. In addition, a significant correlation exists between visual preferences, attention, and emotions [23,24]. Eye tracking technology serves as a common method for measuring individual responses to emotional stimuli. By examining eye tracking focus and attention duration, it is possible to determine the visual preferences of individuals in different settings, as well as the relationship between the public's preferences and landscape [16,24], and landscape complexity [25,26]. The sympathetic response obtained through physiological signals has a higher sensitivity to emotional perception and has been widely applied in existing research on restorative landscapes. However, in existing research, it is often applied to environmental impact studies primarily focused on natural factors. There is relatively little research on high-density urban environments with mixed artificial and natural factors, and there are few studies that combine public eye tracking results in the environment with ECG results to explore the relationship between visual attention and emotion in urban riverside environments.

In densely populated urban areas, rivers meander and converge, representing the most prevalent natural spaces encountered by individuals. The natural environment is more conducive to the recovery of physiological functions than the urban environment [27]. Urban riverside space is the most typical natural space in high-density urban environments and also has a high degree of landscape complexity [28,29]. The urban riverside space, characterized by its complex elements in a high-density urban environment, has emerged as a crucial focal point for addressing a multitude of urban challenges [30]. Therefore, the evaluation and development of riverside landscapes have assumed paramount importance. By examining the impact of different riverside spaces in high-density urban areas through the design and transformation of limited spaces along the riverbanks,

the aim is to move necessary activities in high-density cities closer to spontaneous activities and increase the sociality of urban riverside greenways [31], and achieve optimal emotional regulation among individuals, enhance citizen happiness, and thereby become a central focus of research and design pertaining to high-density urban riverside landscapes. It is worth noting that the public's subjective evaluation of complex scenes tends to be less logical and more closely aligned with aesthetic preferences for similar simple types of landscapes [32]. Existing research in the field of neuroscience primarily focuses on the perception of emotions in relation to green landscapes, blue spaces, and complicated urban landscapes [33,34]. The research on river landscapes is also considered as a whole [28,35], and there is little discussion on the differences among different landscape types in the major categories of riverside landscape. In the context of urban renewal, detailed classification of the same type of landscape and targeted improvement of quality have become the future development direction [36]. Studying the existing riverside landscapes for updating and discovering the impact of different types of riverside landscapes on emotions can provide guidance for the future renovation of riverside landscapes.

The measurement of human emotion in real-life settings present a greater challenge [37]. Previous studies have predominantly employed photographs or videos to simulate scenes, resulting in a passive experiential environment that overlooks the study of human perception regarding boundary landscape components [38]. Nowadays, more and more scholars are using VR devices to collect videos of built environments for simulation experiments, though this method is improved compared to the original method of using photos for experiments [39,40]. However, there is a significant difference in emotional perception between subjects in outdoor environments and virtual reality environments, and outdoor environments have a stronger effect on improving emotions [41]. In the assessment of realistic environments, subjects can effectively alleviate fatigue, enhance cardiopulmonary function, and reduce blood pressure by engaging with natural sounds and inhaling negative oxygen ions in blue and green spaces [42–44]. This, in turn, contributes to the enhancement of the positive impact of riverside spaces on emotions and the reinforcement of public health promotion through everyday landscapes in densely populated cities. The advent of portable wearable sensors has made it feasible to capture the counter experience of natural environments [45]. It can be seen that by collecting the emotional perceptions of individuals in constructed riverside spaces, a more comprehensive and objective evaluation of environmental influence can be achieved, thereby facilitating a more appropriate assessment of spatial design.

The limitations associated with the acquisition of emotional perception, the challenges in identifying subtle emotional differences, and insufficient research on the emotional impact of urban riverside environments are addressed in this study through the utilization of physiological signals derived from ECG and eye movement in real-world environments. These signals are further supplemented by subjective feedback obtained from questionnaires. This research, which relies on the physiological signals of ECG and eye movement, and which is based on small sample experiments, attempts for the first time to evaluate the differences in the impact of different riverside landscape features on emotions in a real environment, explore the relationship between visual attraction features and emotional conversion, and propose suggestions for improving riverside landscape transformation, ultimately enhancing its positive guidance for health and providing scientific guidance.

## 2. Materials and Methods

### 2.1. Research Context and Subjects

This study was conducted in the vicinity of the Liangma River, situated in the bustling Chaoyang District of Beijing, which represents the epitome of urban development with its embassy area, commercial shopping center, and densely populated residential office area. This location serves as a prominent showcase for the capital city's engagement

with the global community, exemplifying a high-density urban setting. The research focused on two river segments located on the northern bank of the Liangma River (Figure 1). The first segment, spanning a length of 500 m, is characterized by towering structures on both sides, with an average channel width of 37 m. The presence of barges and trails in close proximity to the natural environment further accentuates the landscape. The riverbank is strengthened by stone vertical revetments and pine vertical revetments, with slender verdant trees lining the road–river interface. Along the river segment, aquatic plants have been thoughtfully cultivated. The second segment, measuring 270 m in length, features an average channel width of 18 m. It is divided into upper and lower layers, with a height difference of approximately 2 m. A platform green space is interposed between the two layers, while the upper trail runs alongside the railing of the platform green space. The lower trail is situated in close proximity to the river.

Prior to the commencement of the experiment, prospective participants were recruited through an online questionnaire. Only individuals exhibiting emotional stability, physical and mental well-being, non-colored lenses, and access to the designated site were considered eligible for inclusion. The final sample achieved a balanced gender distribution. To ensure the reliability of emotional feedback pertaining to the test, the ECG data collection was conducted in a manner that eliminated the influence of individual momentary fluctuations, thereby bolstering the statistical power of the study despite the relatively small sample size [46,47]. In addition, the dynamic nature of data collection in an outdoor setting posed additional complexities compared to indoor data collection. Therefore, referring to the research of Mohammad et al. [48], a cohort of 13 healthy young adults was selected to participate in the experiment. The data collection took place between 9:00 and 17:00 in November. However, one participant experienced ECG data failure during the data processing stage, leaving a total of 12 individuals with valid data. The male-to-female ratio was 1:1, with an average age of 21 and an average height of 1.69 m.

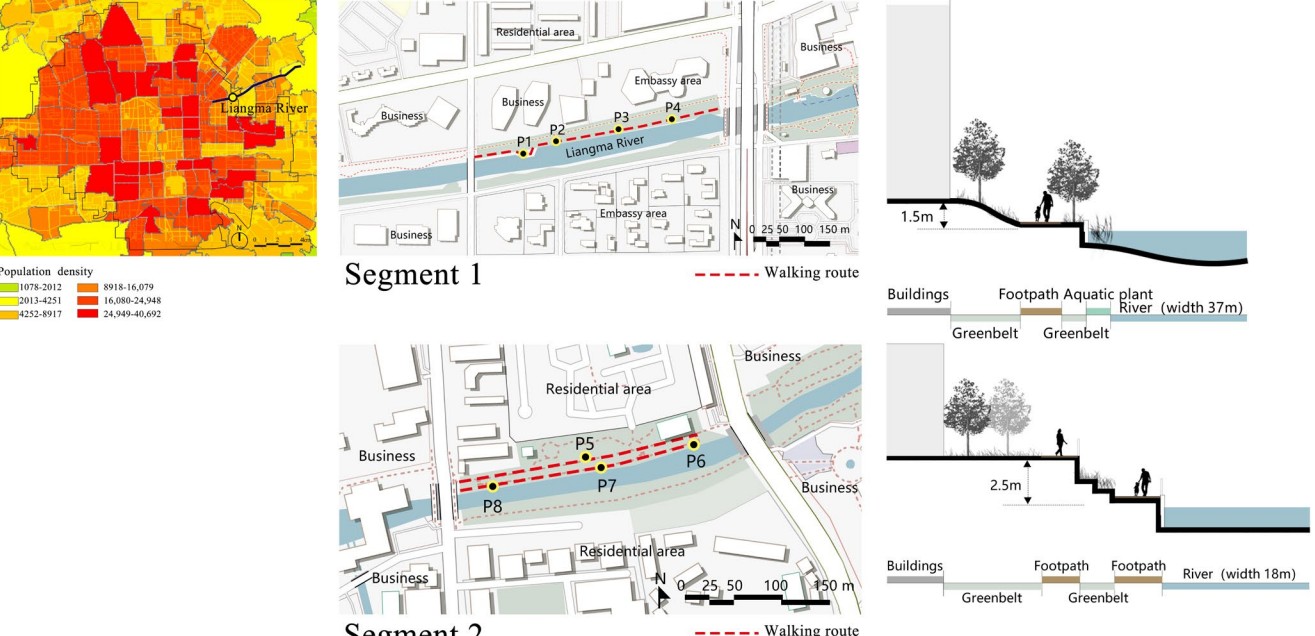

**Figure 1.** Study site and typical river segments.

## 2.2. Experimental Design

The experimental procedure was conducted in accordance with the following protocol (Figure 2): Firstly, the subjects were directed to the vicinity of the experimental site to receive an introduction to the experimental process and necessary precautions. The

subjects' movement trajectory and duration were recorded using the application "two-step path". (1) A portable physiological signal collector (model: Thought Technology-Procomp Infiniti) and portable wearable eye tracker (model: Pupil Labs-pupil invisible) were worn by the subjects. Specifically, three electrodes of the ECG recorder were affixed to the subject's chest at the right margin and intercostal space. Eye movement data was collected as depicted in Figure 3, which showcases photographs of the subjects engaged in the experiment. (2) Physiological data were collected in a high-density urban environment. During a state of calmness, the ECG data were collected for a duration of 2 min prior to entering the segment, serving as the pretest value. (3) Physiological data were collected in an urban riverside environment. Under the guidance of the experimenter, the subjects traversed a distance of 500 m in the riverside environment at an average pace of approximately 1.2 m/s, with channel one representing the upper layer and channel two representing the lower layer. (4) Psychological data were collected. In this study, the PANAS tables developed by Watson were employed to assess mood across 20 items, categorized into two dimensions (Table 1) [49]. Utilizing the Likert 5-point scoring method (1 = not at all, 5 = extremely), dimension scores were assigned, with higher scores indicating a more pronounced emotional state. Following each section of the river course, the experimenter's psychological data were collected through a questionnaire.

**Table 1.** PANAS questionnaire content.

| Positive Emotions (P.A.) | Negative Emotions (N.A.) |
|---|---|
| Interested | Afraid |
| Excited | Jittery |
| Strong | Nervous |
| Enthusiastic | Ashamed |
| Proud | Irritable |
| Alert | Hostile |
| Inspired | Scared |
| Determined | Guilty |
| Attentive | Upset |
| Active | Distressed |

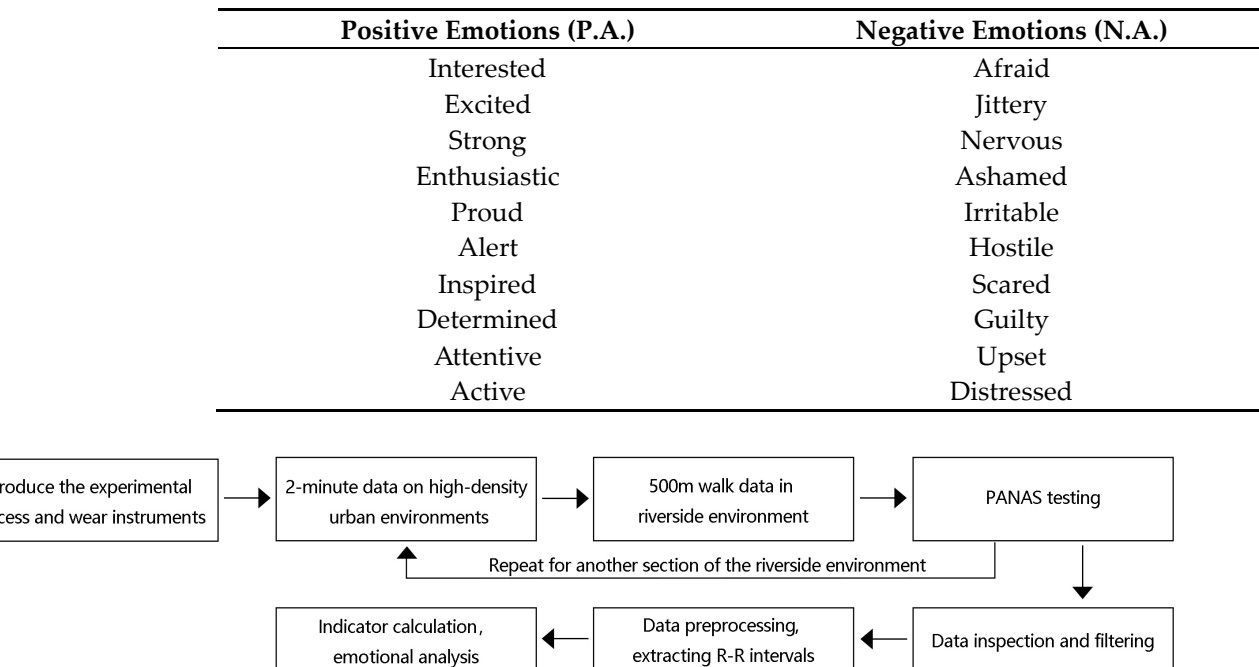

**Figure 2.** Experimental Procedure.

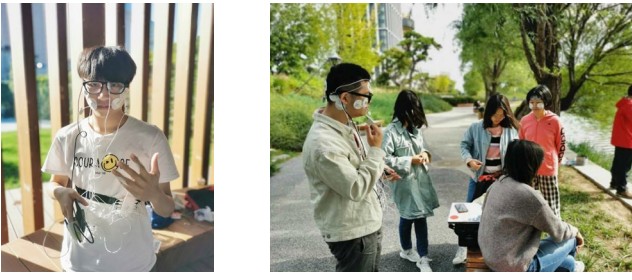

**Figure 3.** Experimental process of subjects.

*2.3. Data Processing*

The two-dimensional emotion model has been widely employed as a prominent low-dimensional emotion model, renowned for its remarkable recognition capabilities based on physiological factors [50]. This model places significant emphasis on the relationship between valence and arousal dimensions in determining emotions. Valence pertains to individuals' subjective evaluation of their environment, with the two extremes being pleasant and unpleasant. On the other hand, arousal reflects the level of activation of bodily energy associated with the emotional state, with the two extremes being excited and calm. By decomposing different emotions into these two dimensions, they can be effectively mapped onto a coordinate system [51]. In terms of the calculated ECG indices, heart rate (HR) corresponds to potency, while high frequency/low frequency (LF/HF) corresponds to arousal degree. Notably, a positive correlation exists between HR and valence, whereas a negative correlation is observed between LF/HF and arousal. In this study, a biofeedback recorder was employed to measure the ECG signals in a live subject environment, with a sampling frequency of 1000 Hz/s (Figure 4).

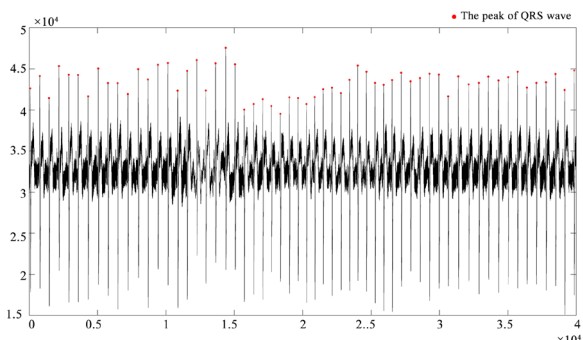

**Figure 4.** ECG signal extraction R-R interval diagram**.**

Following denoising, artifact removal, and normalization pretreatment, the collected ECG signals were subjected to analysis. The heart rate variability (HRV) was found to effectively reflect the mood condition based on the ECG signals, as evidenced by the Lorenz scatter plot. Specifically, SDNN (Equation (1)), RMSSD (Equation (2)), HR, and LF/HF were identified as reliable indicators for emotion recognition. Therefore, an optimal feature segment was selected for data testing. In the Lorenz scatter plot, the long axis (representing the length along the 45° straight line) signifies the overall degree of variation in the inner rate of the test time. Conversely, the short axis (representing the width perpendicular to this straight line) represents the difference in adjacent R.R. intervals, thereby expressing the instantaneous heart rate change and reflecting the activity of the vagus nerve [52]. In a normal heart rhythm, the scatter chart should exhibit a concentrated distribution near the 45° angle, displaying a symmetrical comet-like pattern. However, deviations from the 45° angle indicate changes in mood. Notably, lower values of SDNN and RMSSD correspond to a higher positive influence of the environment on human mood. To further study the relationship between visual preference and emotion regulation, four typical scenes were selected in each section. The position and gaze of eye movement, as well as the attention time within a 10 s interval, were extracted to generate an eye movement hotspot map. Through this analysis, human visual preference was assessed, shedding light on the relationship between visual preference and emotion regulation.

$$SDNN = \sqrt{\frac{\sum_{i=1}^{N}(RR_i - \frac{\sum_{i=1}^{N} RR_i}{N})^2}{N}} \tag{1}$$

$$RMSSD = \frac{\sum_{i=1}^{N} \Delta RR_i{}^2}{N} \qquad (2)$$

where: $RR_i$ means the number of respiration rate, $N$ means the total number of respiration rate.

## 3. Results

### 3.1. Physiological Feedback Results

3.1.1. The Mood Changes of the Riverside Tour

The Lorenz plot in Figure 5 illustrates the significant impact of the riverside environment in high-density cities on the enhancement of residents' mood. The blue data points represent participants residing in a high-density urban environment, while the red data points represent participants in the riverside environment. It is evident that the red data points exhibit a longer spread along the major axis and a wider spread along the minor axis compared to the blue data points. This indicates that in the riverside environment, there is a more pronounced variation in heart rate, a more significant activation of the vagus nerve, and a more intense fluctuation in mood.

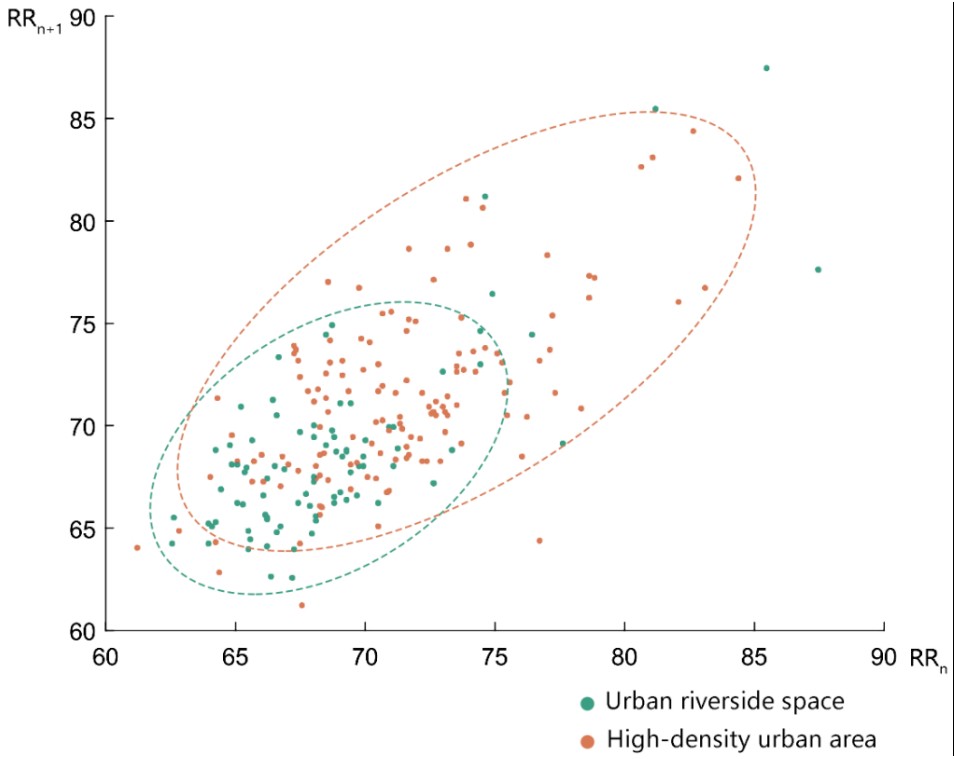

**Figure 5.** Lorenz plot of high-density urban environment and urban riverside environment.

Through a comparative analysis of the effects of two different segments, namely Segment 1 and Segment 2, it becomes apparent that Segment 1 exerts a more significant positive influence on emotional recovery than Segment 2. The emotional states associated with Segment 1 consist of happiness, satisfaction, and relaxation, whereas Segment 2 elicits excitement and happiness. The ECG data successfully passed independent samples t-tests (Table 2). The SDNN, LF/HF, and HR values of Segment 1 and 2 exhibited significant differences at the 0.01 level, while RMSSD demonstrated significance at the 0.05 level. The data comparisons between the two channel types were statistically significant. The Lorenz plot reveals that the overall heart rate in Segment 1 is lower than that in Segment 2 (Figure 6a). The Lorenz plot in Segment 1 tends to be elliptical, with a longer minor axis, whereas the scatter pattern in Segment 2 tends to be tapered, indicating a greater dispersion along

the major axis. These findings suggest that Segment 1 exhibits a more robust positive emotional recovery. Additionally, the experimental results for SDNN and RMSSD in reach 1 were significantly smaller than those in Segment 2. As the pressure is alleviated and emotions develop positively, lower values of SDNN and RMSSD indicate a more pronounced emotional recovery in reach 1 compared to Segment 2. By comparing the data from Segment 1 and 2 (Table 3), it is observed that the average LF/HF index value for Segment 1 is 0.855, which is lower than the corresponding value of 1.246 for Segment 2. This discrepancy can be attributed to the negative correlation between LF/HF and arousal, indicating a lower level of arousal in Segment 2 compared to Segment 1. In addition, the average of HR for Segment 1 is 84.36, which is also lower than the corresponding value of 102.10 for Segment 2. This difference can be attributed to the positive correlation between HR and potency, suggesting a higher level of potency in Segment 2. When projected onto a two-dimensional emotion model (Figure 7), the emotional states associated with Segment 1 tend to be characterized by happiness, satisfaction, and relaxation, whereas Segment 2 elicits excitement and happiness.

The Lorenz plot in each point of Segment 1 is nearly elliptical (Figure 6b), and the degree of dispersion on the long axis direction is relatively low. The emotional changes are relatively stable, and the heart rate gradually increases over time, with the overall emotional trend leaning towards a positive direction. There are more differences in the Lorenz plot in Segment 2 (Figure 6c). The Lorenz plot in point 5 and 6 of Segment 2 are nearly elliptical, with a longer short axis. The dispersion degree of the long axis is relatively low, and emotions tend to be positive. The Lorenz plot in point 7 and 8 are conical, with a higher dispersion degree of the long axis and a greater range of emotional changes. The positive influence of emotions is weakened, with point 8 being the most nervous.

**Table 2.** Independent samples *t*-test.

| | | Levene's-Test | | Mean T-Test | | | | | | | |
|---|---|---|---|---|---|---|---|---|---|---|---|
| | | F | Sig. | t | df | Sig. (2-Tailed) | Mean Deviation | Standard Error | [95% Conf. Interval] | |
| SDNN | Equal Variances Assumed | 0.221 | 0.640 | −3.103 | 248 | 0.003 ** | −7.262 | 2.340 | −11.918 | −2.606 |
| | Equal Variances Not Assumed | | | −3.099 | 241.586 | 0.003 ** | −7.262 | 2.343 | −11.925 | −2.599 |
| RMSSD | Equal Variances Assumed | 0.185 | 0.668 | −2.448 | 248 | 0.017 * | −6.181 | 2.525 | −11.204 | −1.157 |
| | Equal Variances Not Assumed | | | −2.447 | 243.797 | 0.017 * | −6.181 | 2.526 | −11.207 | −1.154 |
| LF/HF | Equal Variances Assumed | 2.488 | 0.119 | −5.568 | 248 | 0.000 ** | −0.391 | 0.070 | −0.531 | −0.251 |
| | Equal Variances Not Assumed | | | −5.542 | 214.529 | 0.000 ** | −0.391 | 0.071 | −0.532 | −0.250 |
| HR | Equal Variances Assumed | 0.067 | 0.797 | −11.375 | 248 | 0.000 ** | −17.740 | 1.560 | −20.843 | −14.637 |
| | Equal Variances Not Assumed | | | −11.387 | 241.091 | 0.000 ** | −17.740 | 1.558 | −20.840 | −14.640 |

** At the level of 0.01, the correlation is significant. * At the level of 0.05, the correlation is significant.

**Table 3.** Data comparison table of Segment 1 and Segment 2.

| Index | Group | N | Mean | Std. Err. | Std. Dev. |
|---|---|---|---|---|---|
| SDNN | 1 | 126 | 32.595 | 10.146 | 1.566 |
| | 2 | 124 | 39.857 | 11.161 | 1.743 |
| RMSSD | 1 | 126 | 26.944 | 11.234 | 1.733 |
| | 2 | 124 | 33.124 | 11.765 | 1.837 |
| LF/HF | 1 | 126 | 0.855 | 0.253 | 0.039 |
| | 2 | 124 | 1.246 | 0.376 | 0.059 |
| HR | 1 | 126 | 84.36 | 7.397 | 1.141 |
| | 2 | 124 | 102.10 | 6.789 | 1.060 |

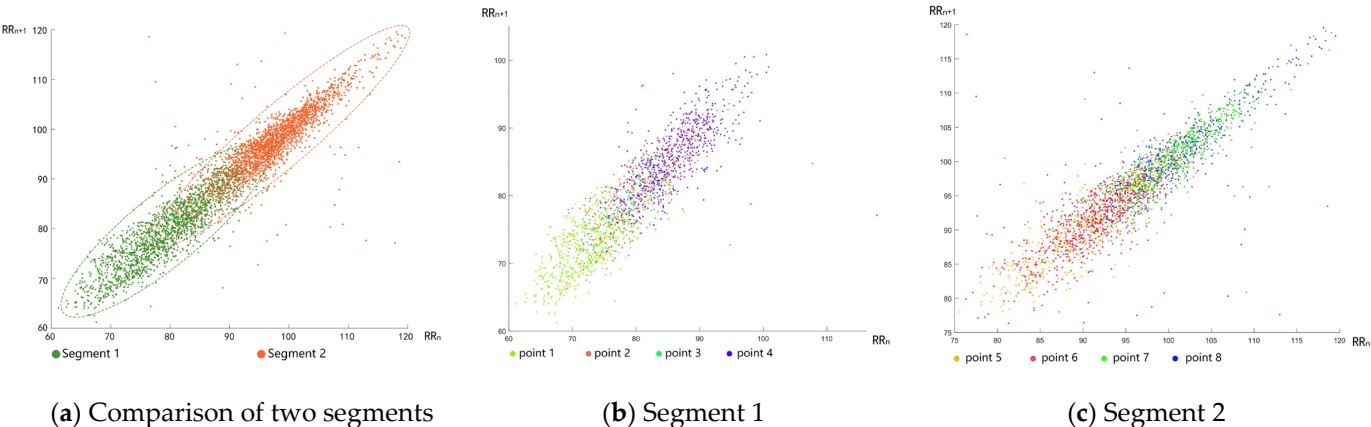

(**a**) Comparison of two segments     (**b**) Segment 1     (**c**) Segment 2

**Figure 6.** Comparison of Lorenz plots in two segments and each segment point.

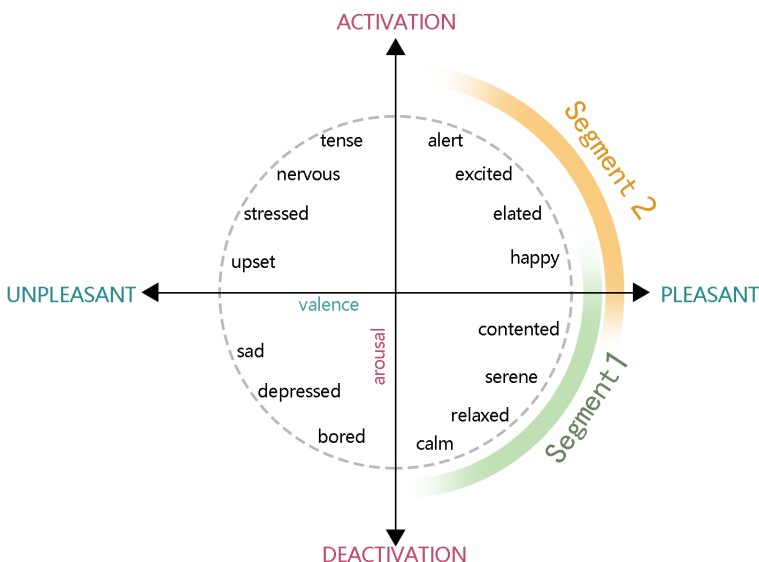

**Figure 7.** Emotion corresponding to two segments of river in two-dimensional emotion model.

### 3.1.2. Visual Attraction of the Riverside Scene

The analysis of the eight river scenes reveals several observations: (1) The populace exhibits a lower focus on complicated urban elements such as buildings and retaining walls, instead directing their attention towards the natural surroundings including plants and water bodies. (2) In scenes characterized by extensive riverside greening in densely populated cities (points 1, 3, 4, 5, and 7), the visual emphasis tends to gravitate towards the waterfront side. Specifically, the line of sight predominantly fixates on aquatic plants, while the greenery near the city side holds less attraction. (3) In environments featuring a high degree of hardening along the trail (points 7 and 8), green plants exert a greater pull on the line of sight. (4) The dispersion of people's gaze along the riverside footpath is directly proportional to the openness of the surrounding area. This dispersion serves as a reflection of individuals' appreciation and preference for the environment. Notably, pedestrians exhibit a more scattered gaze and greater visual perception in naturalistic scenes (points 3, 4, and 5) and scenes with designated rest spaces (point 8)—though planting trees on the waterfront side can hinder the view of the water surface (point 2)—as opposed to complex scenes where pedestrians primarily focus on the surrounding landscape, points with a high degree of hardening, placing greater emphasis on walking and displaying less appreciation for the surrounding landscape.

*3.2. Feedback Results of the Questionnaire*

Concerning the PANAS score line graph depicting different subjects (Figure 8), it becomes evident that, considering the limited sample size, subjects harbor diverse psychological sentiments towards various types of river landscapes, lacking any apparent tendency. Upon examining the mean comparison of each assessment item (Figure 9), it is observed that if the value of "interested" and "strong" in channel 1 is less than that in channel 2, and if the value of any other term in channel 1 is significantly greater than that in channel 2, four items ("excited", "inspired", "determined", and "active") exhibit relatively significant changes. Conversely, in terms of negative emotions, no difference is observed between the values of "distressed" and "irritable" in the two channels. However, the value of "afraid" surpasses that of channel 2, while the value of any other term in channel 2 exceeds that of channel 1. Notably, two items ("scared" and "nervous") demonstrate relatively significant numerical fluctuations.

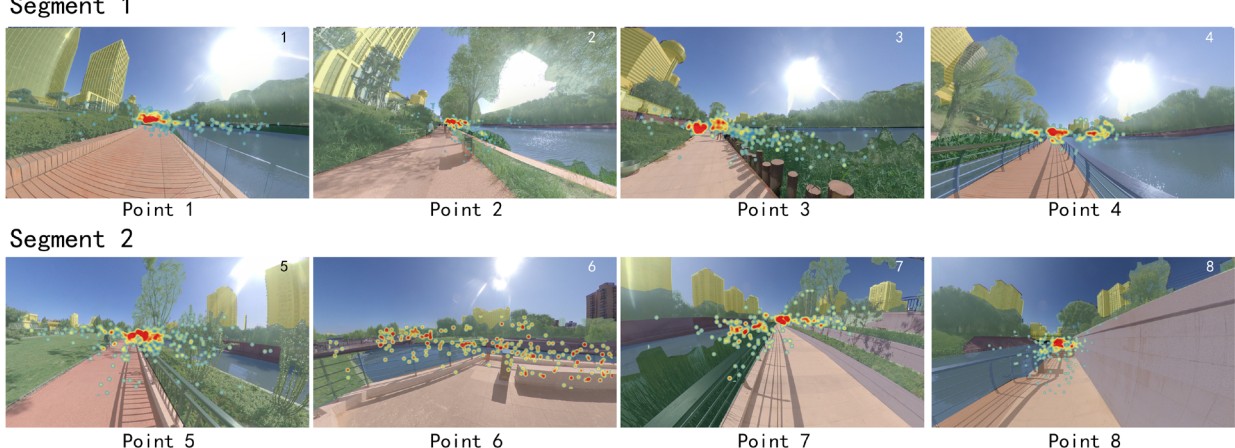

**Figure 8.** Heat map of eye movement data for each point.

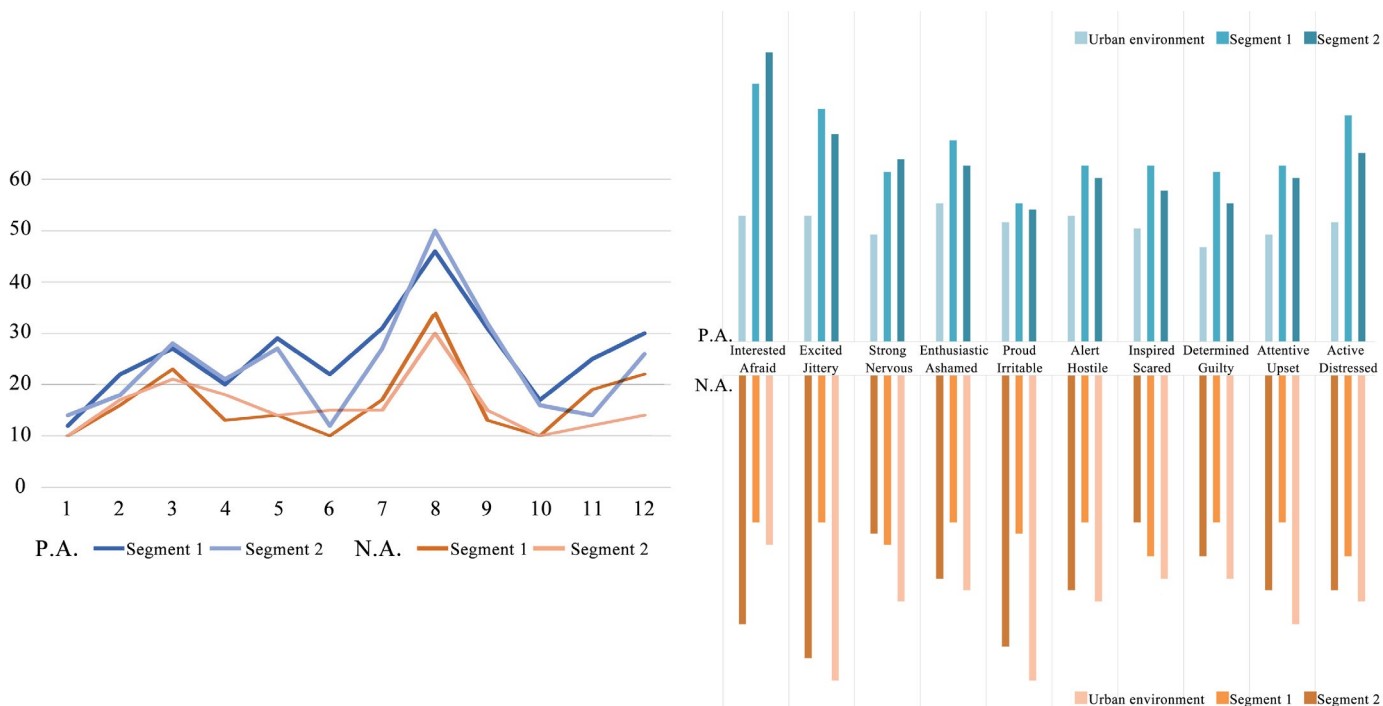

**Figure 9.** Subjective psychological perception of the subjects.

## 4. Discussion and Analysis

### 4.1. The Influence of High-Density Urban Riverside Landscape on Mood

Existing research indicates that urban riverside environments serve as a green haven, offering urban dwellers a space for daily strolls and activities [53], thereby facilitating emotional engagement with the environment and contributing to the overall well-being of the populace. In contrast to the demanding urban setting, the composite landscape comprising the blue and green spaces in the high-density urban river channel exerts a positive influence on emotions [54], with varying effects depending on the composition of the riverside spaces. Here, we show that areas adorned with fewer artificial structures and abundant vegetation, including plants and aquatic flora, exhibit superior stress-relieving properties. Conversely, riverside spaces characterized by extensive hardening tend to disperse the sightlines of pedestrians seeking respite, as they direct their gaze towards the riverside landscape, with a preference for open spaces along the waterfront.

In a more naturalized riverside environment, it has a sustained enhancing effect on positive emotions. In the hardened riverside environment, the positive impact on emotions shows a trend of first increasing and then decreasing.

Building upon the findings, which highlight the pivotal role of plants in capturing visual attention [55], we corroborate the significance of aquatic plants and shrubs in the riverside trail scene. Notably, individuals display greater attentiveness towards aquatic plants, surpassing that directed towards shrubs. Therefore, the intensity of plant presence in the riverside trail environment can be ranked as follows: aquatic plants > shrubs > water. This result highlights the scarcity and value of outdoor water as a precious natural element in the urban environment [56]. By incorporating aquatic plants, the urban river, which is inherently challenging, assumes a more visually natural appearance, enriches the water environment, enhances the overall landscape quality, and significantly alleviates stress.

### 4.2. Analysis of the Mechanism of Different Types of Riverside Landscapes on Emotions

The emotional impact of urban riverside landscapes, characterized by a high degree of naturalization and a significant proportion of artificial elements, varies significantly. The underlying factors contributing to these differences are as follows:

(1) The visual perception and emotional relief effect of the riverside environment are related to the level of separation between pedestrians and the densely populated urban surroundings. By employing the D/H theory in the analysis of street spaces, the distance (D) between the waterfront environment and the buildings, as well as the height (H) of the buildings, are taken into consideration. In instances where the D/H ratio of the high-density urban waterfront space is <1, a sense of urgency is established, and the complicated landscape adversely affects vision [57]. To mitigate this, the introduction of tall green vegetation between the waterfront city buildings and the waterfront footpath facilitates a green spatial transition, thereby reducing the perceived pressure exerted by the buildings and optimizing the regulatory effect of the waterfront environment on emotions. In addition, the difference in elevation between the riverside space and the high-density urban environment creates a physical barrier, temporarily detaching individuals from the urban setting and providing a respite from the associated pressures and moods [58]. While the level of rigidity in the riverside activity space surpasses that of other trail scenes, varying spatial attributes yield different feedback results. The waterfront activity area offers individuals the opportunity to engage with water and serves as an interactive platform, thereby eliciting a positive emotional impact. Notably, the water surface significantly influences emotional recovery [59]. Due to its high openness and pronounced visual perception, the waterfront activity area exhibits a superior emotional recovery effect.

(2) The difference in the proportion of green space to water bodies in the blue and green space was examined. Existing research indicates that the level of exposure to blue and green space exhibited a strong correlation with individuals' subjective perception of

happiness [60]. By incorporating a higher level of greenery and increased water coverage, the secretion of adrenaline and the activation of sympathetic nerves were mitigated to a certain extent, resulting in a more pleasant and relaxed mood [61]. Therefore, this configuration provided the most rejuvenating landscape. In this study, the width of channel 1 was approximately twice that of channel 2, thereby affording a broader and more expansive visual perception of water. Moreover, the elevation difference between the waterfront space and the urban area adjacent to river channel 1 exhibited a gentle slope. The presence of dense vegetation along both sides of the riverside trail contributed to an exceptional green visual rate and sound attenuation effects. Conversely, the upper space of river channel 2 exerted a specific shading effect, while the lower space featured a complex landscape predominantly characterized by shrubs, a low green visual rate, and heavy artificial traces, thereby easily producing a sense of excitement.

(3) The transition between the riverside space and the urban interface varied in terms of the mode of transition, and the positive emotional promotion effect of visual perception is related to the distance between pedestrians and hard landscapes. Plant communities were found to induce a more calming effect, whereas buildings elicited a more stimulating response [62]. In Segment 1, there is a gentle slope green space separating the waterfront space from the urban interface, which effectively shielded the high-rise structures from view when traversing the riverside trail. Therefore, the visibility of urban development was significantly reduced. In Segment 2, the lower riverside space and the urban interface were separated by a rigid retaining wall, with tall urban buildings situated on the opposite side of the river. As a result, individuals could identify the artificial traces of construction in the riverside space. It is evident that the incorporation of trees, gradual slopes, and vertical greening between waterfront spaces and high-rise buildings can foster a more relaxed and tranquil emotional state.

Under normal walking conditions, people mainly look straight ahead. In Segment 1, there is a height difference between the city and the riverside trail, and plants divide the space between the city and the riverside trail; the impact of architecture on visual perception is relatively small. In Segment 2, the hard retaining wall is tightly attached to the walkway; the artificial landscape with low attention has a stronger impact on pedestrian vision. This situation leads to concentrated vision, low attention to the riverside landscape, and a reduced restorative effect.

(4) Different types of revetments were observed. The difference in height between the riverside trail and the average water level was minimal. A narrow green space flanked the water's edge, featuring stone vertical revetments or pine revetments, with aquatic plants serving as transitional elements. This configuration exhibited a greater tendency towards natural aesthetics, thereby fostering a sense of calm and relaxation. In contrast, river 2 was divided into two tiers of walking systems, with an approximate height difference of 2 m. The lower trail featured vertical revetments and railings, thereby accentuating the artificial nature of the surroundings. Therefore, emotional arousal was higher, resulting in a greater propensity for excitement.

*4.3. Determination of Real-Scene Emotion Perception under the New Technology*

This study drew on the research methodology of quantifying physiological perception in the field of human engineering technology. It integrated this approach with a psychological questionnaire to address the limitations of conventional subjective questionnaire evaluations. The adaptation of emotions to the environment in most scenarios is subjective to an individual's perspective [63]. When data collection is not convenient in real-world environments, the stability of small sample psychological subjective perception is weaker compared to physiological feedback. Conversely, the ECG data, obtained from a large sample size, were not influenced by subjective evaluation errors, thereby highlighting the potential of human cause engineering technology in obtaining accurate emotional evaluations even with limited sample sizes.

The ECG results revealed that the riverside environment exerted a positive impact on individuals' emotions, with a greater natural ambiance leading to increased feelings of happiness, satisfaction, and relaxation. Conversely, a more artificial riverside setting evoked greater excitement and happiness. The analysis of physiological perception data and the evaluation results from the psychological questionnaire demonstrated consistency. This study presents a novel avenue for interdisciplinary research in landscape architecture, human cause engineering, and other engineering technology disciplines. It introduces a feedback-based evaluation of the influence of landscapes on individuals in landscape planning and design, fostering enhanced interaction between humans and their surroundings. Moreover, it offers a compelling real-time method for evaluating landscapes in natural environments. For instance, by perceiving the emotional feedback effects and influence mechanisms of individuals in riverside environments, landscape design can guide emotional changes based on the expected emotional values of people. This approach proves instrumental in enhancing the impact of landscape planning and design on public health guidance.

### 4.4. Riverfront Landscape Planning and Design Recommendations

The positive influence exerted by the urban riverside landscape on emotions presents a remarkable opportunity for enhancing the well-being of residents in densely populated urban areas. The waterfront region in high-density cities represents a highly diverse and complex ecosystem [64], serving as a primary spatial medium for promoting human health. In the context of a densely populated city, the design of riverside sites, aided by human engineering technology, facilitates the regulation of public health by considering the collective mood and site requirements. This involves pre-construction environmental adjustments and post-construction emotional regulation, as well as local modifications and enhancements based on physiological feedback. By bolstering the waterfront environment, public health gains can be achieved, thereby meeting the residential needs and acting as a catalyst for physical and mental well-being among the surrounding population, thus realizing the objective of establishing a "healthy city".

In planning and design, using environmental psychology, environment, and behavior as theoretical foundations and exploring the activity patterns of people in outdoor landscape environments can help better develop urban spaces [32]. During the planning phase, it is necessary to fully exploit the potential of the riverside space to enhance connectivity and accessibility between the waterfront area and the city, thereby enabling a greater number of residents to benefit from the convenience and health-enhancing attributes of the urban river space [53]. In the design of the riverside landscape, particular emphasis is placed on strengthening the visual perception of the water surface and ensuring the visibility of the riverside periphery (Figure 10a). In addition, the construction of a diverse range of riverside spaces, including natural walking paths and areas for recreational activities, is undertaken in response to the emotional recuperation requirements associated with the riverside landscape.

To strengthen the presence of the water environment and establish a serene riverside landscape, an overhanging horizontal promenade, positioned in close proximity to the water surface, was incorporated to enhance the visual attraction and perception of the aquatic body (Figure 10f). While adhering to the principle of non-intrusion into the river channel's width, a hydrophilic area and an aquatic vegetation planting strip were introduced onto the rigid riverside revetment. This measure aimed to enhance the visibility and verdant panorama of the riverside's azure expanse. In addition, the foliage density on the urban side of the riverside footpath was enriched, employing trees with lofty branching patterns to offer shade for the riverside trails. This arrangement also served to mitigate the visual impact exerted by urban buildings upon the riverside space. Adjacent to the waterfront trail, a verdant space was designated, predominantly characterized by low-lying grass structures or grass structures that do not obstruct the line of sight (Figure 10e). In close proximity to the water's surface on the waterfront side, an appropriate

increase of aquatic plantings was implemented to improve the canopy level of vegetation and enhance the verdant panorama of the canopy. This approach aimed to fully exploit the innate beauty of plant communities while attenuating the presence of urban artificiality (Figure 10d). The implementation of a gentle slope revetment, composed of wooden piers and stones, was explored as a potential substitute for the riverside fence. This alternative was pursued to ensure the safety of riverside recreational activities while mitigating the perception of artificiality. In conjunction with the riverside trail's robust continuity and profound vistas, the introduction of an open canopy facilitated an enhanced perception of the natural horizon and improved the dynamism of the vista (Figure 10b).

To create enthusiastic and exciting riverside landscapes in planning and design, low-growing herbs were strategically planted along the waterfront, while a grass structure was established on the urban side. To alleviate plant density, trees were positioned away from the trail. Additionally, the integration of revetment design facilitated controlled flooding of the site. Artificial structures, such as seating areas and sports facilities, were thoughtfully arranged to maximize the social potential of the site and foster social interaction among residents (Figure 10c).

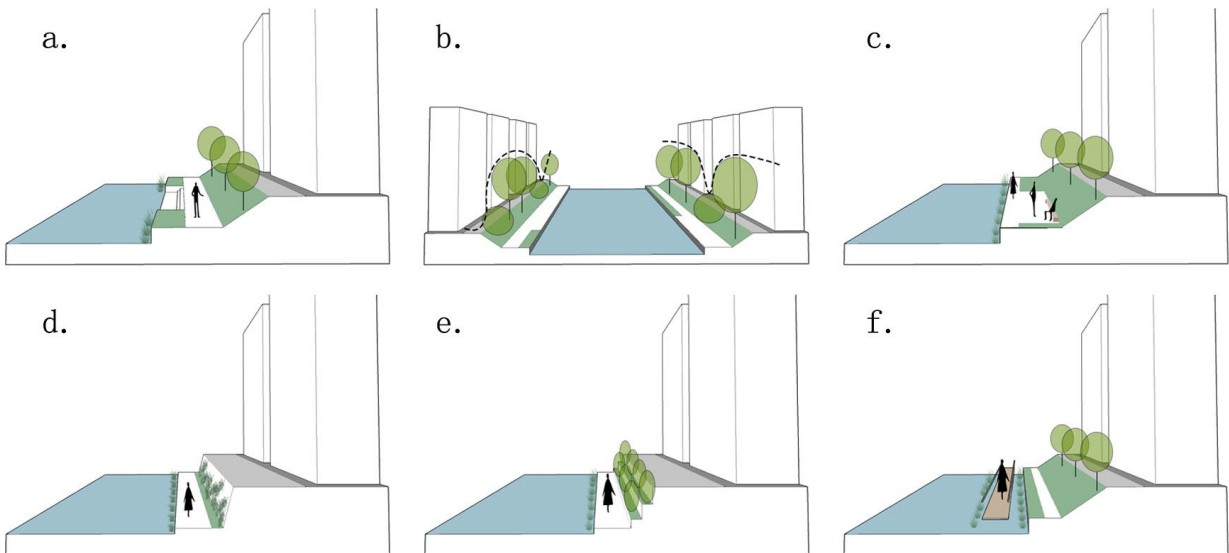

**Figure 10.** Suggestions for improvement of riverside landscape renovation. (**a**) Planting trees on the city side reduces the visual impact of urban architecture; (**b**) planting trees and shrubs, constructing undulating canopy lines and skylines (the dotted line represents the canopy line); (**c**) constructing waterfront spaces for activities and rest; (**d**) vertical greening with height difference weakens the artificial hard landscape; (**e**) treatment of elevation difference as terrace landscape; (**f**) design water trails to enhance interaction with water.

## 5. Conclusions

Drawing upon small sample empirical research conducted in the field, this paper delves into the utilization of ECG feedback and questionnaire feedback as a means to measure the impact of high-density heterogeneous urban river real-scene environments on individuals' mood and health. By conducting both physiological and psychological assessments, we preliminarily substantiated the following three aspects: (1) urban river landscapes have a positive influence on people's mood and health. In addition, we have discovered that different riverside landscapes possess varying emotional guidance effects, with emotional changes being closely linked to the components of the riverside landscape. People tend to be happy and relaxed in riverside environments with a high proportion of natural elements, while those in areas with a high degree of artificial riverside environments tend to be excited and happy. A more naturalized riverside trail environment has a stronger sustained positive effect on emotions. (2) When traveling along the riverside trail, people tend to prefer scenes with open and high spaces, meanwhile paying the

highest attention to aquatic plants. (3) In real-life environments where large-scale experiments are not convenient, the results of small sample experiments obtained through physiological feedback are more convincing than survey questionnaires. However, due to the limited sample size, further development is needed in this research. Based on the current research, designers can design the elements of the site to enhance the experiential environment and amplify the specific emotional guidance effect of the environment, thereby enhancing the health promotion effect of the landscape.

However, this study exhibits several limitations that warrant further study. The absence of a separate quantitative analysis of the river landscape components introduces uncertainty regarding the degree of influence exerted by each component. In addition, there remains a lack of quantitative analysis pertaining to the emotional improvement benefits of the riverside environment subsequent to its transformation and enhancement. In future studies, control variables can be further explored by means of scenario simulation, allowing for a comparative analysis of emotional improvement effects before and after modification. This will enable a more comprehensive understanding of the significance of each element in the riverside scene in relation to emotional improvement.

**Author Contributions:** Conceptualization, X.L. and M.L.; methodology, M.L. and D.W.; software, X.L.; validation, R.L.; formal analysis, X.L.; investigation, M.L.; resources, D.W.; data curation, R.L.; writing—original draft preparation, R.L. and M.L.; writing—review and editing, X.L., S.Z. and M.L.; visualization, D.W.; supervision, X.L. and S.Z. All authors have read and agreed to the published version of the manuscript.

**Funding:** This research supported by the "Youth Project of Beijing Social Science Foundation" (19SRC012).

**Institutional Review Board Statement:** Not applicable.

**Informed Consent Statement:** Not applicable.

**Data Availability Statement:** Data is contained within the article.

**Acknowledgments:** The authors are indebted to the reviewers and editors for their valuable comments and suggestions.

**Conflicts of Interest:** The authors declare no conflicts of interest.

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
