# Peer review of "The Effect of Perceived Real-Scene Environment of a River in a High-Density Urban Area on Emotions"

_land, doi:10.3390/land13010035_

Round 1
Reviewer 1 Report
Comments and Suggestions for Authors
The topic and the idea of the study is very interesting and valuable, also corresponds very well to the scope and aims of the Special Issue. Generally, most stages of the study are well developed, however there are some weaknesses related to the selected aspects of the presented material.
Main comments and suggestions for Authors:
1. The title is clear. Key words are well selected and in line with the topic.
2. The Abstract is well organized and includes main results of the study.
3. The Introduction is well developed and Authors mention most important aspects related to conducted study, what create wide background. The current state of the research field has been presented.
The aim of the study is not clearly formulated – it has rather descriptive form and main studied aspects are listed in continuous way with some arguments/explanation (lines: 133-147). The aim must be easy to understand thus more briefly mentioned, making it easy to connect with conclusions.
4. Section 2. Material and Methods has appropriate order and includes most important elements. The sample of respondents is low and the limitations related to respondents’ selection /participation should be more argued in my opinion. At the same time this qualifies the presented research more as a pilot (or preliminary) study, what also should be clearly stated in the text – both in the introductory part explaining the main aim, as well as in the section of Material and Methods.
The used methods including data processing stage are quite deeply presented step by step, what explains clearly how the study was conducted.
5. The section of Results includes main and important information and is supported by rather short but enough comments, also graphs and figures.
Figure 9 has wrong number.
6. Discussion follows main aspects studied by Authors, but at the same time it is quite extensive in relation to the results of such a small research sample, which is surprising. Regardless of this approach, some more relations to other studies (references) should be added to increase the value of presented research.
7. Conclusions – they are rather too brief in my opinion. At the same time the Authors should be more careful when formulating conclusions. Also the phrase such as e.g. "different riverside landscapes possess varying emotional guidance effects", is too strong. Due to the small research sample, it is better to point out that e.g. the preliminary (pilot) study has shown that different riverside landscapes may possess varying emotional guidance effects, but the study needs further development, etc.
Summing up, a clear indication that the study is preliminary (pilot) should be included in description, also the main aim should be more clearly presented. Also conclusions needs some revision.
Others:
- there are missed spaces between words and brackets with reference number in the whole text, etc. – it must be revised.
Author Response
Dear Reviewers,
Thanks very much for taking your time to review our manuscript entitled “The effect of perceived real-scene environment of a river in a high-density urban area on emotions” (Manuscript ID: land - 2752234). We would like to take this opportunity to submit our revised manuscript to land.
We are so grateful for the valuable comments and suggestions from the reviewers made on the previous version of our manuscript. The manuscript has been revised according to these thoughtful comments. The revised content is changed to blue in the manuscript.
In this letter, we provide our responses, point-by-point, to the comments and suggestions indicated by the reviewers.
- Summing up, a clear indication that the study is preliminary (pilot) should be included in description, also the main aim should be more clearly presented. Also conclusions needs some revision.
Response: Thanks for your suggestions.
- We described in the introduction the sample size of participants in previous physiological feedback data experiments (lines 75-78).
In comparison to traditional questionnaire surveys, the utilization of physiological signals for emotion recognition is not susceptible to subjective influences, thereby enabling accurate research results even with a limited number of participants [21,22].
And add experimental references for selecting a small number of samples in the methods section of this article (line 173).
referring to the research of Mohammad et al. [23], a cohort of 13 healthy young adults was selected to participate in the experiment.
- We have rewritten the research objectives and summarized them. (Line 140-145).
This research relies on the physiological signals of ECG and eye movement, and based on small sample experiments, attempts for the first time to evaluate the differences in the impact of different riverside landscape features on emotions in a real environment, explore the relationship between visual attraction features and emotional conversion, and propose suggestions for improving riverside landscape transformation, enhancing its positive guidance for health and providing scientific guidance.
- In the research objectives and results, it is indicated that this experiment is a preliminary study and further in-depth research is needed. More detailed explanations of the experimental results have been provided (lines 140-141, lines 520-536).
Drawing upon small sample empirical research conducted in the field, this paper delves into the utilization of ECG feedback and questionnaire feedback as means to measure the impact of high-density heterogeneous urban river real scene environments on individuals' mood and health. By conducting both physiological and psychological assessments, we preliminarily substantiated the following three aspects: 1) urban river landscapes have a positive influence on people's mood and health. In addition, we have discovered that different riverside landscapes possess varying emotional guidance effects, with emotional changes being closely linked to the components of the riverside landscape. People tend to be happy and relaxed in riverside environments with a high proportion of natural elements, while those with a high degree of artificial riverside environments tend to be excited and happy. A more naturalized riverside trail environment has a stronger sustained positive effect on emotions. 2) When traveling along the riverside trail, people tend to prefer scenes with open and high spaces, meanwhile pay the highest attention to aquatic plants. 3) In real-life environments where large-scale experiments are not convenient, the results of small sample experiments obtained through physiological feedback are more convincing than survey questionnaires. However, due to limited sample size, further development is needed in this research. Based on the current research, designers can design the elements of the site to enhance the experiential environment and amplify the specific emotional guidance effect of the environment, thereby enhancing the health promotion effect of the landscape.
- Add other research (references) as support in the discussion section.
- Number error in Figure 9
Response: Thanks for your suggestions. We have changed the numbering of Figures 9 and 10.
- there are missed spaces between words and brackets with reference number in the whole text, etc.
Response: Thanks for your suggestions. We have made modifications to the formatting of the references cited throughout the entire manuscript.
We made a considerable effort to improve the manuscript and made many changes in the revised manuscript based on the comments and concerns raised by the editor and reviewers. Thanks very much again for giving us the opportunity to revise this manuscript. We greatly appreciate the editor and reviewers for their help in improving the overall clarity and quality of the work in this manuscript. We sincerely hope that the revised manuscript will meet with your approval and look forward to hearing from you.
Sincerely yours,
Xin Li
Reviewer 2 Report
Comments and Suggestions for Authors
please see attachment

Author Response
Dear Reviewers,
Thanks very much for taking your time to review our manuscript entitled “The effect of perceived real-scene environment of a river in a high-density urban area on emotions” (Manuscript ID: land - 2752234). We would like to take this opportunity to submit our revised manuscript to land.
We are so grateful for the valuable comments and suggestions from the reviewers made on the previous version of our manuscript. The manuscript has been revised according to these thoughtful comments. The revised content is changed to blue in the manuscript.
In this letter, we provide our responses, point-by-point, to the comments and suggestions indicated by the reviewers.
Reviewer #2:
- I am curious what results the authors would obtain during the same tests, but conducted in the evening or at night. Certainly, some of these places will seem dangerous to users. But other boulevards - thanks to beautiful lighting and the reflection of light in the water will be judged as more beautiful.
Response: Thanks for your suggestions. Unfortunately, the experiment was not conducted at different time periods and was conducted during the day. Subsequent research will supplement the perception of night scenes and compare the results between day and night.
- Authors should add doi.
Response: Thanks for your suggestions. We have added doi at the end of the references.
- I would like to recommend to the authors the research of Danish psychologist and urban planner Jan Gehl.
Response: Thanks for your recommend. (I referred to Jan Gehl's research and cited it in the article, please refer to the line 101-102 and 469-471 for details.
We made a considerable effort to improve the manuscript and made many changes in the revised manuscript based on the comments and concerns raised by the editor and reviewers. Thanks very much again for giving us the opportunity to revise this manuscript. We greatly appreciate the editor and reviewers for their help in improving the overall clarity and quality of the work in this manuscript. We sincerely hope that the revised manuscript will meet with your approval and look forward to hearing from you.
Sincerely yours,
Xin Li

Round 2
Reviewer 1 Report
Comments and Suggestions for Authors
I appreciate all works done by the Authors, main suggestions has been introduced.
The presentation of the aim of the study has been improved – it is clearly formulated. The information that the study is preliminary and based on small sample has been introduced. Discussion, and especially conclusions, are clear.
I can recommend publication of the manuscript in its present form.
Author Response
Thank you to the experts for the acceptance and affirmation.